# Algorithmic Typewriter Art: Can 1000 Words Paint a Picture?

Jules Kuehn*          David Mould†

Carleton University

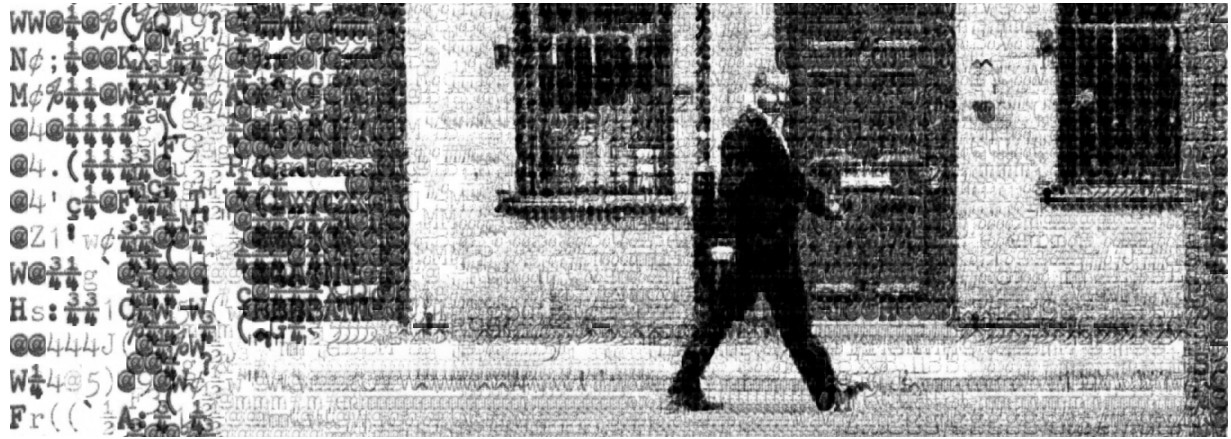

Figure 1: The left side shows one layer of typed characters. Moving right, more overlapping layers are added.

## ABSTRACT

We present an optimization-based algorithm for converting input photographs into typewriter art. Taking advantage of the typist's ability to move the paper in the typewriter, the optimization algorithm selects characters for four overlapping, staggered layers of type. By typing the characters as instructed, the typist can reproduce the image on the typewriter.

Compared to text-mode ASCII art, allowing characters to overlap greatly increases tonal range and spatial resolution, at the expense of exponentially increasing the search space. We use a simulated annealing search to find an approximate solution in this high-dimensional search space. Considering only one dimension at a time, we measure the effect of changing a single character in the simulated typed result, repeatedly iterating over all the characters composing the image.

Both simulated and physical typed results have a high degree of detail, while still being clearly recognizable as type art. The accuracy of the physical typed result is largely limited by human error and the mechanics of the typewriter.

**Index Terms:** K.6.2—Computer Graphics—Non-photorealistic rendering; K.6.3—Computer Graphics—Image Processing

## 1 INTRODUCTION

*Typewriter art* involves producing images with typewritten text. A modern computer graphics practitioner is likely familiar with ASCII art, where an image is formed out of text characters on the screen. Typewriter art offers additional flexibility, insofar as characters are not restricted to a non-overlapping grid. Multiple characters can be typed at a single location, a practice called *overstriking*, and offset rows and columns of characters can partially overlap with previously typed rows and columns.

---

*e-mail: jules.kuehn@carleton.ca
†e-mail: mould@scs.carleton.ca

Furthermore, keys on a mechanical typewriter can be struck with varying levels of force, transferring greater or lesser quantities of ink from the typewriter ribbon. Varying the strike force produces a much smoother tonal range than monochrome ASCII art, which is especially important in lighter-tone regions of the image. Overstriking and overlapping improves outcomes in the darker regions of the image and increases detail.

In this paper, we present an algorithm for converting an input image into typewriter art, exploiting overstriking, overlapping and strike force to add detail and to improve tone matching. We can directly render a simulated typed image, or produce a set of instructions that can be typed to create a physical realization of the image, in keeping with recent trends in computer graphics towards assisting computational fabrication [2].

Manually crafted typewriter art can be extremely detailed, and historically, it was often created using primarily the period key or other small, geometric shapes. Without restriction to a regular grid, the technique of overlapping small characters yields both fine spatial resolution and a perceptually wide tonal range, with a texture reminiscent of pointillism or stippling.

Considerable artistic skill was needed to craft these works. However, the typewriter also allowed users with less skill to produce images. "Typewriter mystery games" [17] provided typists with instructions that, when carried out, produced an image. These instructions exploited the backspace key to enable overstriking to create much darker shades.

In our method, the instructions are simply rendered type for four separate layers, each offset by half a character horizontally, vertically, or both. Formally, the four layers have their respective origins at (0,0), (0.5,0), (0, 0.5), and (0.5,0.5) × (charWidth, charHeight). Together, overstriking and half-spacing provide nearly full ink coverage. Figure 1 shows a rendered result, where superimposed layers of text cooperate to form a detailed image.

As outlined in Figure 3, our process takes as inputs a target image to be reproduced on the typewriter and a scan of the typewriter's character set. The program selects the characters to be typed for each of the four layers, which then overlap to produce the image. Within each layer, the placement of characters is limited to a grid dictated

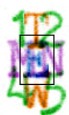

Figure 2: Nine offset characters overlap within one character position

by the typewriter itself. The algorithm optimizes by measuring the effect of changing a single character in the simulated typed result, repeatedly iterating over all character positions. By typing the selected characters for each layer, we can reproduce the image on the typewriter without requiring any artistic skill.

With respect to both tone reproduction and shape matching, this technique produces a better approximation of the target image than non-overlapping ASCII art with a similar number of characters. Exact reproduction, however, is not the goal; if we were to find a pixel-perfect reproduction, the charm of text art would be obscured. The aesthetic effect relies on some degree of difference remaining between the target image and its re-creation through overlapping characters.

Whimsically, we posed the question "Can 1000 words paint a picture?" The answer is "yes": 5790 characters suffice for a fair facsimile of a portrait, though complex scenes require more resolution.

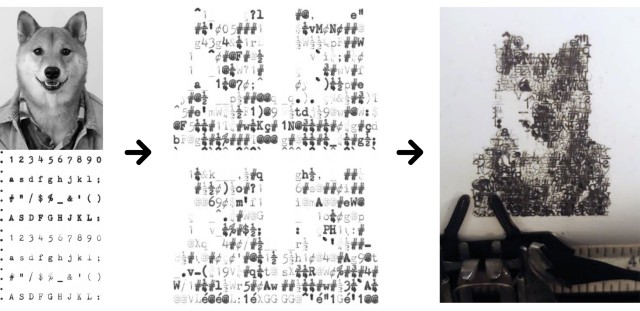

Figure 3: Left to right: input photo and character set, generated instructions for layers, typed result

This paper makes the following contributions:

- We exploit overlapping characters to increase the dynamic range and expressiveness of text art.

- We present an optimization algorithm for automatic creation of typewriter art, paying special attention to physical reproducibility on a mechanical typewriter.

- We propose *asymmetric mean squared error*, in which positive error (too little ink) is weighted less than negative error (too much ink). Subjectively, this produced the best results.

## 2 BACKGROUND

Much recent work in ASCII art has focused on improving shape matching, which can be traced back to the introduction of the Structural Similarity metric (SSIM) [24]. Another metric, created specifically for ASCII art, is the Alignment Insensitive Structural Similarity Metric (AISS) [21]; Xu et al. deform the target image to better match the available character shapes at a given position. Deforming the target image poses issues for the high-fidelity approach we pursue, but the idea of optimizing the alignment of the target image proves useful.

Conventionally, ASCII art used monospaced fonts. Xu et al. [22, 23] achieved superior results using proportional-width fonts.

Although this approach provides flexibility in the columns of type, the rows of type are still fixed; typewriter art allows both to vary.

Some recent approaches to generating ASCII art involve machine learning with no explicit metric. Akiyama employed a convolutional neural network, trained on manually created structural ASCII art, to produce compelling results for this style [1]. Markus et al. use a decision tree to approximate SSIM comparison for a particular character set, yielding a good approximation at high speed [16]. With our interest in overlapping characters, these trained model approaches pose the issue that only single characters are stored in the model, so overlapping composites of those characters would not be compared with the target image.

In typewriter art, the characters can in principle be freely positioned, which brings to mind stippling [5]. Computer-generated stippling usually seeks non-overlapping stipples, unlike our case which encourages overlaps. Also, while stippling need not be restricted to points [4, 11], computer-generated stippling methods do not exploit choice of object shape to better approximate an input image, whereas this is a fundamental aspect of ASCII art and typewriter art. Stroke-based rendering allows overlapping strokes; Hertzmann's painterly rendering method [10] tracks the difference between the current canvas and a target image, adding new strokes where the error is largest, akin to our strategy of choosing the character for a given cell so as to minimize the difference with the input photograph.

## 3 ALGORITHM

The aim of this project was to marry the mechanical reproducibility of the "typewriter mystery game" with the improved spatial resolution of freehand typewriter art by employing four overlapping layers, offset both vertically and horizontally. We simulated the effect of overlapping characters with multiplicative compositing.

### 3.1 Preparing the character set

Our algorithm requires as inputs a target image to be reproduced, an image of a "character set" , the number of rows and columns in the character set, and the desired number of rows in the typed output. The character set image must be cropped into individual images of characters. To ensure integer crop boundaries – and integer character placement at half spacing – the width of the character set image must be divisible by the number of rows in the character set times two, and so on for height. The image is first minimally stretched in each dimension until it matches these constraints, and then it is sliced into individual characters, removing all but one blank.

This process may distort the aspect ratio of the characters slightly. To ensure physical reproducibility, the algorithm applies the same aspect distortions to the target image, then uniformly scales it to (character width) × (desired row length). When optimization is complete, the simulated typed result is inversely transformed to the correct character aspect ratio.

### 3.2 Search technique

We take an iterative approach, selecting a character for a single position at a time. A greedy selection answers the question "What character, placed in this position, will best complement the already-chosen characters to most closely match the target image?" After a selection for each character has been made, a single optimization cycle is complete.

Due to the use of overlapping layers, past selections may later become suboptimal: when any character overlapping a position changes, that position must be re-evaluated. For example, if all neighbours of a certain position changed from a dark character to a light one, the selection for that position should possibly be changed to a darker one to compensate. Moreover, because all the characters are connected through overlap, changing a single selection can trigger a cascade of selection changes spanning the image. To

ensure termination, we limit the search to a maximum number of optimization cycles.

The algorithm incorporates overstrike (placing multiple characters at the same position) by using the simulated result produced by one run of the program as a background layer during compositing operations in a second run.

### 3.2.1 Simulated annealing

A greedy best-first search evaluates every position along a single dimension, selecting the character that results in the highest similarity to the target image. However, there is no guarantee that this is optimal in other dimensions. The greedy search is quick to converge to a local optimum, where no single character swap will increase the similarity to the target image.

To combat the lock-in to local optima exhibited by a greedy search strategy, we use stochastic simulated annealing (SA) to intelligently expand the search space [8].

Each time the algorithm visits a character location, it evaluates the candidate characters in a random order. If substituting the candidate character for the existing character at this position reduces loss, the candidate is selected. A candidate that increases loss may also be selected, with probability inversely proportional to the delta between the current loss and the loss resulting from the candidate's selection.

---

**Algorithm 1:** Optimization algorithm with SA

Create list of positions [charPos]: {layer, row, col, charId};
Initialize each charId in [charPos] randomly;
**for** $i \leftarrow 1$ **to** *maxIterations* **do**
   **for** $pos \in$ shuffled(*charPos*) **do**
      Compute curLoss with currently selected character*;*
      **for** *candidateId* $\in$ shuffled(*charIds*) **do**
         Compute newLoss with candidate character*;*
         $delta \leftarrow curLoss - newLoss;$
         $p \leftarrow delta/temperature;$
         **if** $delta > 0$ *or* $e^p >$ random$(0,1)$ **then**
            $pos.charId \leftarrow candidateId;$
            *break;*
      $temperature \leftarrow temperature - coolingStep;$
      **if** *temperature* $\leq 0$ **then**
         $temperature \leftarrow initTemp \times reheat;$
         $initTemp \leftarrow temperature;$

---

The width of this probability distribution is decreased over iterations, as the temperature is reduced after each selection. When the temperature reaches 0, it reheats to a fraction of the initial temperature. We empirically chose an initial temperature of 1e-2, a cooling step of 1e-4, and a reheating factor of 1e-1.

### 3.3 Loss function

In Algorithm 1, loss is computed by comparing the simulated, composite text within the bounds of a character position to the corresponding area of the target image to be approximated.

We relied on previous work in image fidelity measurement to provide loss functions used to score character selections. We also employed a variation on mean squared error (MSE), *asymmetric mean squared error* (AMSE), which multiplies positive error by a factor of $1 + a$ before squaring.

$$AMSE = \left(\frac{1}{n}\right) \sum_{i=1}^{n} ((y_i - x_i) \times k_i)^2$$

$$k_i = \begin{cases} 1 + a, & \text{if } y_i - x_i > 0 \\ 1, & \text{otherwise} \end{cases}$$

While asymmetric loss functions are well studied in machine learning [7, 9, 20], their application to image stylization is, to our knowledge, novel.

We also investigated a combined loss function $(1 - ssim) \times amse$ and maximizing SSIM; see Section 5.1.

### 3.4 Optimized cropping

In our method, characters can only be placed on a fixed "character grid". To improve alignment of features in the target image with possible features given by the overlapping characters, we first generate quick approximations – using a single, greedy optimization cycle – for each of 64 slightly different crops of the target image. The cropping that maximizes $SSIM \times 4 + PSNR$ is applied to the input before iterative optimization.

Whereas the iterative optimization visits positions in a random order, the greedy optimization applied here uses a priority order. The order in which a position is visited is determined by the maximal decrease in loss resulting from selecting a new character at that position.

The set of crop parameters is all combinations of translation parameters (x, y) and scale parameter s:
$x = -a \times charWidth$
$y = -b \times charHeight$
$s = (c + n)/n$; n = number of characters per row in layer (0,0)
$a, b, c \in \{0, 0.25, 0.5, 0.75\}$

## 4 RESULTS

Unless otherwise indicated, all type images are simulated. We use the notation [20w] to indicate a result generated at 20 characters in width with 1 pass (no overstrike), and [20w, $X$p] to indicate 20 characters in width with $X$ passes.

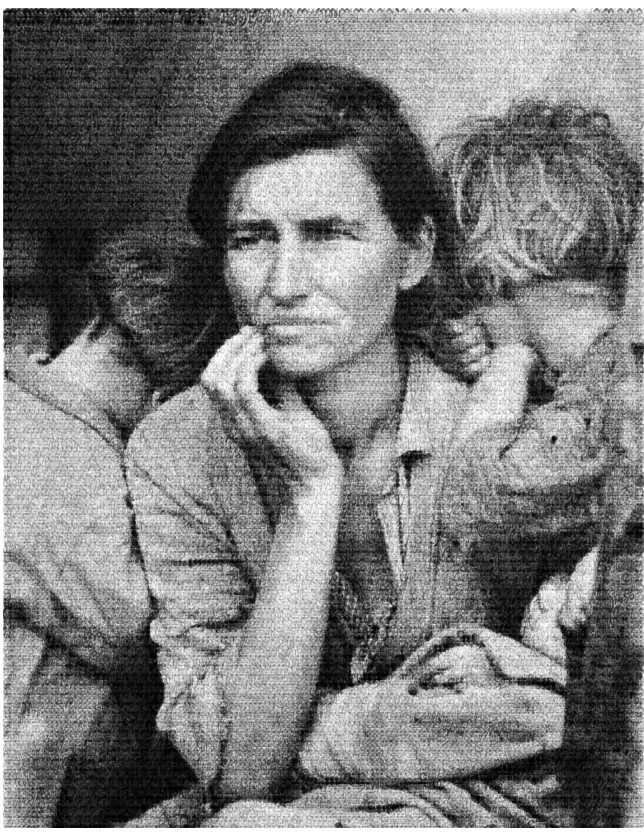

Figure 4: "Migrant mother", optimized settings [68w,3p]

Figure 4 shows a high resolution (68 characters wide) simulated typed image; if physically typed, the image would be 5" x 8". At this resolution, the algorithm reproduces a target image including multiple persons with excellent quality, conveying the emotional impact of the input photograph [13].

Figure 5 shows 8 images produced with the same settings. To best answer the question posed by the title of this paper - can 1000 words paint a picture - we limit ourselves to only 5790 typed characters. This number reflects the average English word length (4.79) plus one space per word [18]. Each image was sized to maximally use our character budget across two passes. AMSE is the loss function, with asymmetry settings of $a = 1.5$ on the first pass, and $a = 0.1$ on the second. In general, the simulated typed results communicate the content and mood of the input photographs regardless of content.

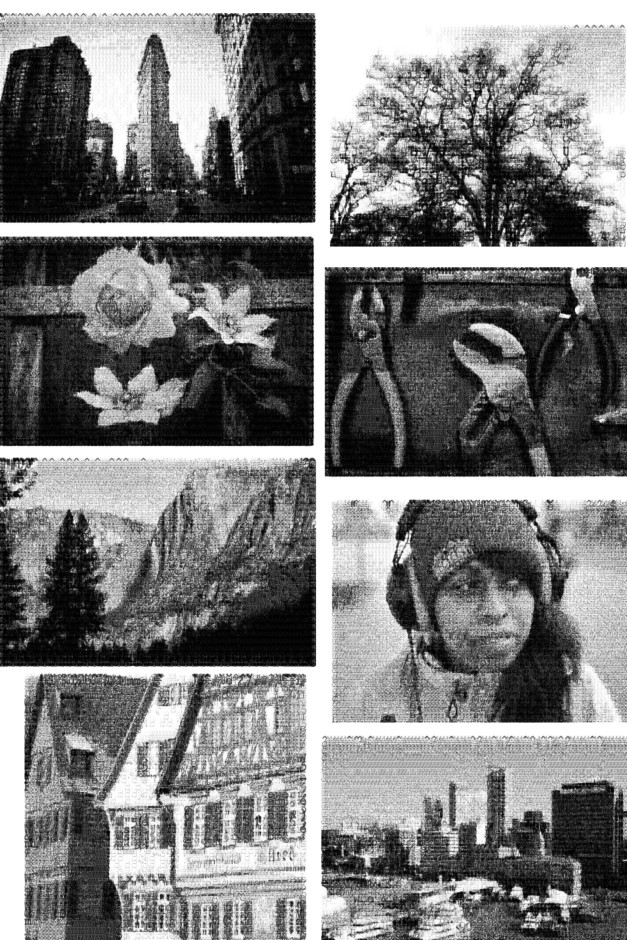

Figure 5: Gallery of results with consistent settings [ 40w,2p]. Zoom to see details.

To represent noisy textures, the algorithm selects a variety of overlapping characters, demonstrated in the thin tree branches, or the windows of the Flatiron Building (top left). In the absence of texture, characters with more even ink distribution – such as @ or # – are selected. This applies to both light and dark regions, with the darkness of the character and number of overlaps determining overall tone. This is evident in the skin tones of the woman with headphones and in the sky of several images, as well as the gradient shown in Figure 15.

Employing overstrike achieves good tonal range in the shadows, as seen in the handles of the wrenches and the windows of darker buildings. It can also produce a deep black level, demonstrated in the landscape and flowers results. Further emphasizing the value of overstrike, light shapes against a dark background are especially clear, like the round nail suspending the rightmost wrench. While the fixed character grid is set at 1/2 character resolution, some details in the output demonstrate a spatial resolution of less than 1/16 of a character.

In general, individual characters are difficult to discern except in the lightest areas, where characters do not overlap – e.g., the sky in the cityscape. The algorithm selects characters that nest, hiding their textual origins, unless the character shapes are a particularly good match – as in the underscores used for the cruise ship decks.

Details with low local contrast, such as the face of the woman with headphones, display coarser spatial resolution than areas of high local contrast, such as the edges of buildings. This is due to relying on overlapping characters to produce sharp edges. When additional overlap would cause a poorer tonal match, shape resolution suffers. This explains why metrics preferring shape matching result in higher local contrast.

However, relying on MSE suits a high-fidelity approach. Accurate tone matching makes the shadows in the cityscape and waterfront images easy to discern; the three-dimensional nature of the buildings is clearly conveyed. Employing a shape-matching metric such as SSIM obscures this.

### 4.1 Physical typed results

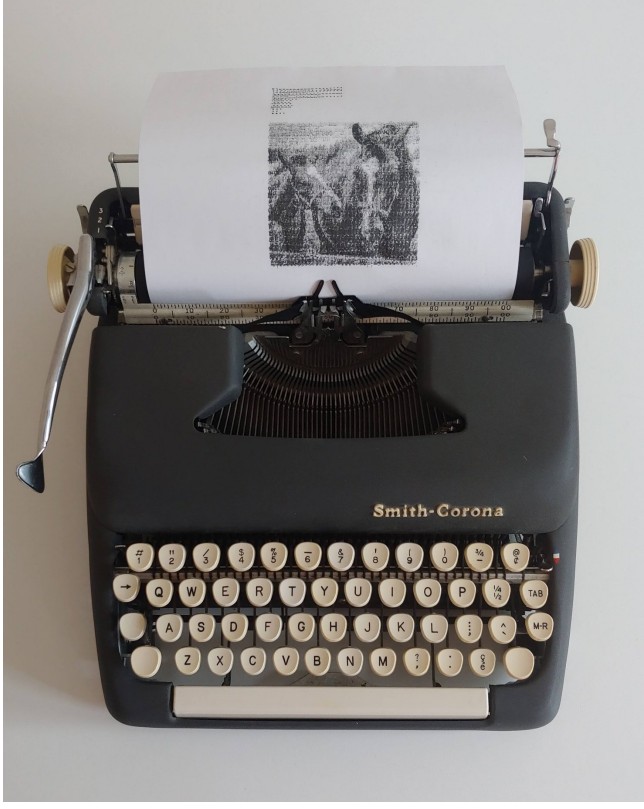

Figure 6: Physical typed result, in typewriter [40w,2p]

Physical reproducibility on a mechanical typewriter is a key feature of this work. Figure 6 shows the result of following the generated instructions to type a 70 cm$^2$ image over 6 hours.

Physically typing the images revealed difficulties in replicating the simulated results. The physical typed result in Figure 7 shows false textures and loss of blackness in the body of the horses, especially in the bottom right quadrant. Even slight misalignment of the

offset layers creates gaps between tightly packed characters – especially noticeable when thin characters like $\frac{1}{2}$ overlap within areas of uniform tone. This is an example of over-optimization: while packing thin characters can improve outcomes in the simulation, in practice the physical result would be improved by favouring wider, overlapping characters in smooth-textured regions.

Tonal relationships in the simulated result are reflected in the physical typed result, albeit with a compressed dynamic range. The loss of blackness results both from alignment errors and the incongruity between the subtractive colour space of typed ink and the additive colour space of the simulation. The simulation composes layers of ink using the approximation of multiplicative compositing. In reality, the combination of ink and paper limits the black level, while inconsistent strike force creates a cruder step between the lightest reproducible tones and the white paper background.

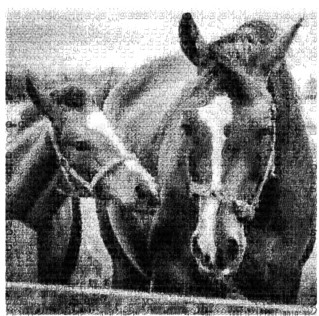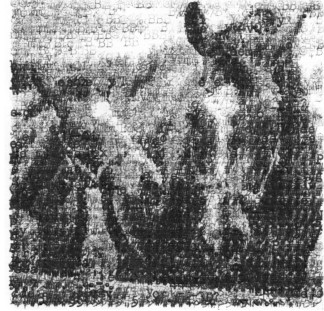

Figure 7: Left: Simulated; Right: Physically typed [40w,2p]

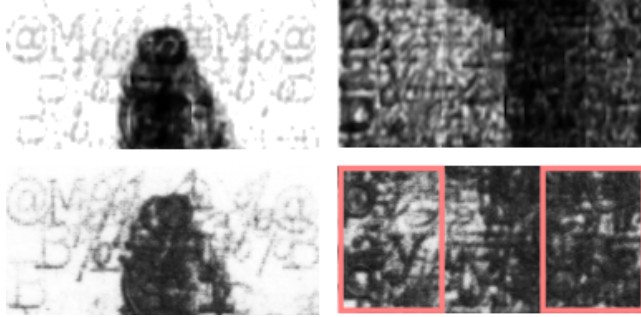

Figure 8: Detail of Figure 7. Top: Simulated; Bottom: Physically typed

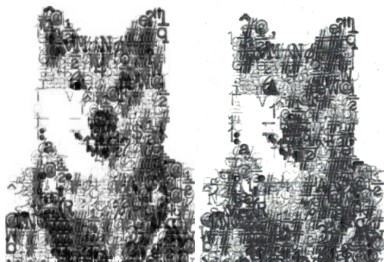

Figure 9: Left: Simulated; Right: Physically typed [16w,1p]

Figure 8 shows two details from Figure 7. The left images – detail of the top-most rows – show excellent alignment of the characters. Tone matching is accurate overall. In contrast, the right images – detail of lower rows – demonstrate substantial misalignment. While

the layers are properly aligned at the top of the image, moving the page to create an offset layer can introduce unwanted rotation. Here, the page was rotated counter-clockwise while typing the layer containing $\frac{1}{2}$ and y, relative to the layer containing the leftmost b characters (left highlight). Thus, $\frac{1}{2}$ and y are shifted to the right relative to the b characters. Due to the algorithm's reliance on precise placement of the thin $\frac{1}{2}$ characters, this misalignment also creates gaps in what should be a uniform dark area (right highlight).

While the typed result of the horses has few misplaced characters (we counted 10 / 6400), achieving such a result is difficult. Figure 9 shows the high visibility of a misplaced character in areas of high local contrast such as the dog's left ear.

Unlike the horses image, Figure 9 employs only one pass, without additional overstrike; as a result, the reduction in contrast from rendered to physical result is less dramatic than in Figure 7. The optimization of subsequent, overstrike passes depends heavily on the algorithm's flawed multiplicative compositing, so the simulation deviates further from physical reality as more overstrike is allowed.

## 5 DISCUSSION

In this section, we discuss various elements and choices that can affect our results. We preview the main findings here; additional details can be found in the following subsections.

The loss function used for comparison between the target image and the simulated typed result has, subjectively, a strong effect on the results. Best results were obtained from an asymmetric variation on MSE that disproportionately penalizes wrongly placed ink. Blending this metric with SSIM can sharpen results, especially when the typed image is small.

Greedy search will necessarily converge to a local optimum, in which no single character swap can improve the score. We can get closer to the overall global optimum state by using simulated annealing.

Alignment of the target image with the typewriter's "character grid" can have a strong effect on shape matching. Alignment can be optimized by measuring the effect of translation and scaling on the target image, selecting the best parameters for each. This is the primary factor when the typed image is small.

Allowing multiple characters to be typed at a single position can substantially increase tonal range and shape matching.

Using a character set that contains variation in strike force is the largest factor in obtaining a good tonal range in the midtones and highlights.

### 5.1 Loss function

In this application, MSE matches tone quality well. The penalty for larger error at a single pixel gives it a sharper, shape matching quality than mean absolute error.

AMSE penalizes positive error (too much ink) to a greater degree than negative error (too little ink). This has two advantages: first, it minimizes erroneously placed ink, which is subjectively more noticeable; second, an overlapping character can fill in missing ink later, while erroneously placed ink cannot be removed. In practice, AMSE improves shape matching – both subjectively and as measured by SSIM on the resultant image – while only slightly compromising MSE.

As a loss function SSIM differed markedly from MSE in this application, working much like an edge detector with little respect for tone matching. At small sizes, SSIM can yield a linear, sketch-like style, and some of the best results come from blending SSIM with AMSE. The loss function $(1 - SSIM) \times AMSE$ performs well at sizes of around 20 characters (Figure 10), slightly emphasizing lines while leaving some lighter areas bare of ink. Effectively, this increases sharpness and contrast.

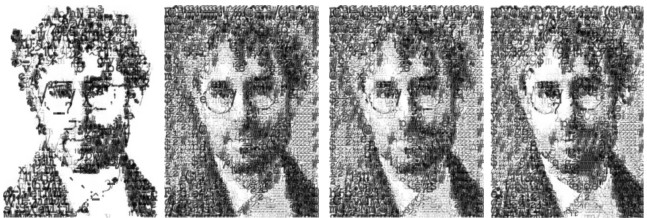

Figure 10: Left to right: SSIM, MSE, AMSE ($a = 0.2$), (1-SSIM $\times$ AMSE ($a = 0.2$) [20w]

## 5.2 Search technique

### 5.2.1 Sensitivity to initial state

To evaluate the effectiveness of greedy search, we performed many runs of the algorithm from different random states; figure 11 shows representative examples. The exhibited sensitivity to initial state indicates that the local optimum reached by a greedy search may still be some distance from the global optimum.

The results using SA from different random states (Figure 11) are hard to distinguish. All SA results captured the whites of the eyes while some greedy results blurred them out. SA also consistently increased SSIM and PSNR across all trials.

**Greedy**

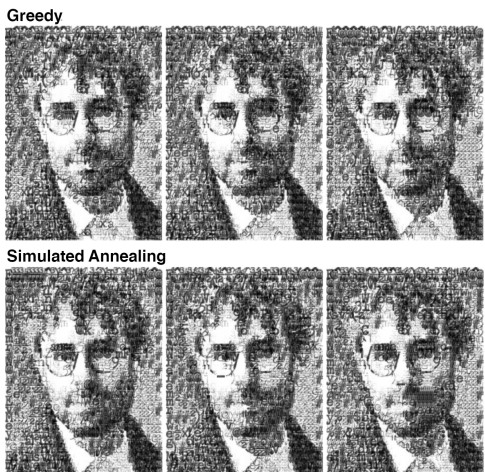

**Simulated Annealing**

Figure 11: Greedy and simulated annealing results from random initial states [20w]

### 5.2.2 Selection order

The order in which selections are made has a predictable effect on the output, especially evident when the search is initiated from a blank state. The first selections will tend to be overly dark, as the single character under evaluation takes full responsibility for tone matching over an area that could ultimately include 9 overlapping characters; see Figure 2. A greedy search tends to become trapped in local optima with overly dark initial selections.

We experimented with priority-ordered selection, but we found that it does not improve the results when simulated annealing is used, and comes at a high computational cost. Employing random selection order and random initial state was sufficient to avoid these issues without incurring high computational cost, and hence we took this approach for all our results.

## 5.3 Pre-processing the target image

Since many source images are colour, the black and white conversion method has a substantial effect on the result (Figure 12) [12, 15]. We considered these processes out of scope, but we have worked to optimize one critical aspect of image pre-processing unique to this application: the alignment of the target image to the typewriter's "character grid" .

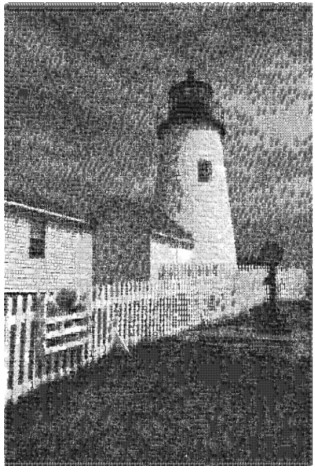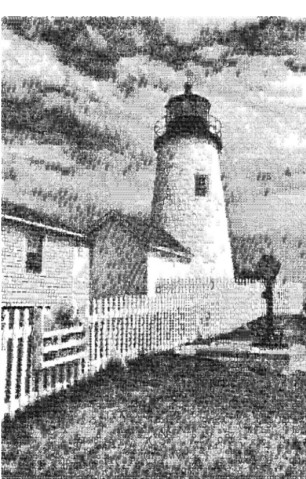

Figure 12: Left: RGB average; Right: STRESS algorithm [40w]

### 5.3.1 Optimized cropping (auto-alignment)

Figure 13 shows two results for the same checkered test pattern, with and without cropping (scaling and shifting the image within the same fixed-size container). The results after cropping via the auto-alignment routine are much sharper. In realistic scenarios – rendering a portrait, say – the alignment still has a strong effect.

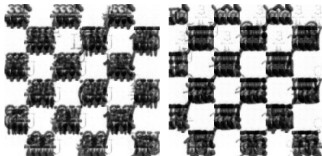

Figure 13: Effect of optimized cropping on test pattern. Left: no crop; Right: optimized crop [12w]

In the right panel of Figure 14, the target image was slightly scaled and shifted to better align with the "character grid" of the typed result. Note the improvement in shape resolution of the eyes, glasses, hair and nose. At small image sizes, alignment is one of the dominant factors in the quality of the result.

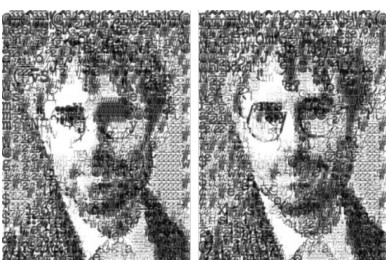

Figure 14: Effect of optimized cropping on portrait. Left: No crop; Right: Optimized crop [20w]

## 5.4 Number of overstrike characters at a single position and number of total characters

Allowing the typist to "overstrike" multiple characters at the same position within the same layer increases the tonal range – especially allowing darker tones – and can to some extent increase shape matching (Figure 15, Figure 16).

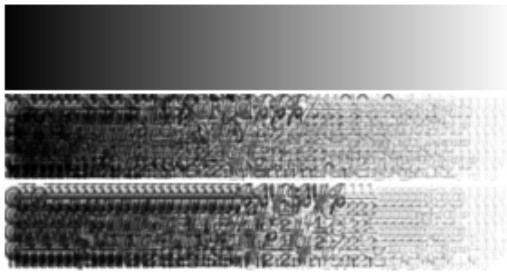

Figure 15: From top to bottom: Target gradient, result with 2 passes, result with 1 pass [20w]

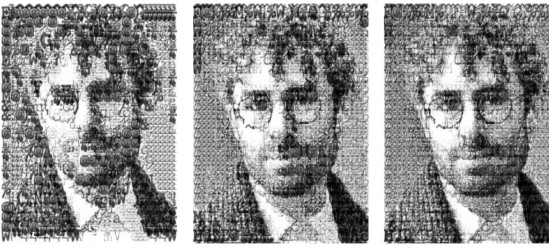

Figure 16: Distributing ink between multiple passes with overstrike. Left to right: 1, 2 and 3 passes [19w]

In exploring selection order, we learned that it was preferable to distribute ink between several layers, rather than concentrating ink in a few. By default, adding a second, overstrike pass will contribute little ink, as the majority of the ink has already been placed in the first pass. With the AMSE metric, we can increase the penalty for wrongly placed ink (asymmetry) on the first pass and reduce it on the second pass. A higher asymmetry factor encourages the selection of characters that are an especially good match in some areas, but have too little ink in others, over characters that better match the tone but do not match the shape as closely. A subsequent pass, with a lower asymmetry factor, can fill in the gaps.

Unsurprisingly, increasing character resolution produces better results. Figure 17 shows two simulations, each with three passes, at 20 characters wide and 30 characters wide. The higher resolution yielded a marked improvement in detail.

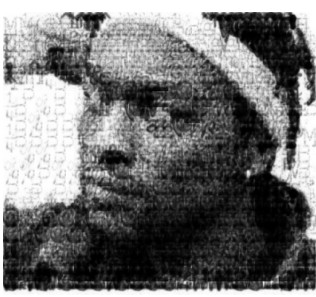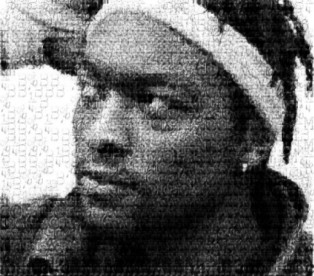

Figure 17: Left: [20w,3p]; Right: [30w,3p]

Keeping to a fixed maximum number of characters, generally the best results come from distributing the characters between two passes. Darker images especially benefit from overstrike, as seen in Figure 18. The increased tonal range in the shadows more than compensates for the reduction in size up to around four passes.

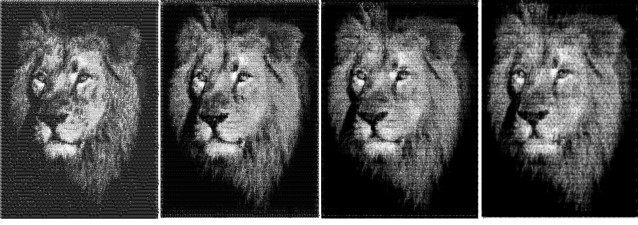

Figure 18: Left to right: [45w,1p], [32w,2p], [23w,4p], [16w,8p]

## 5.5 Character set

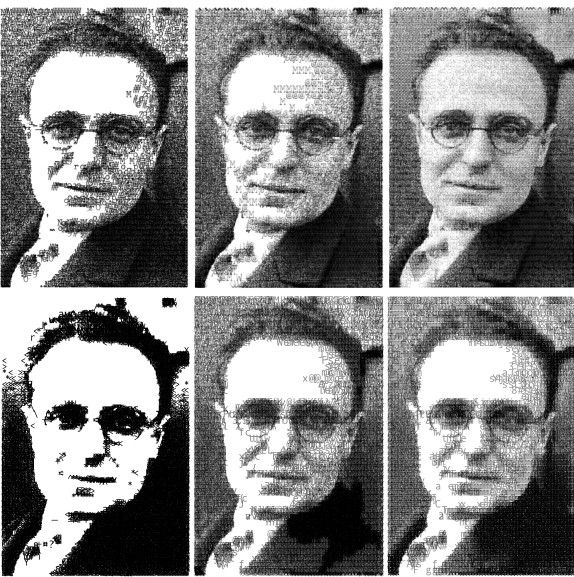

Figure 19: Characters from Smith-Corona typewriter with dry ribbon, typed hard and soft

The choice of typewriter greatly influences the outcome, as it forms the character set that is available to the algorithm. With most mechanical typewriters, the typist can vary their strike force to produce lighter and darker variations of the same character. This has dramatic results: including lighter variations greatly increases tonal range in the midtones.

Figure 20: Top: Typed characters; Bottom: SF Mono font; Left to right: 1 tone, 2 tones, 2 lighter tones [30w,4p]

Figure 20 shows the same target image rendered with different character sets, including a monospace computer font. The leftmost images, created with monotone character sets, exhibit limited tonal

range and draw more attention to the textual characters. In comparing these two images alone, the font does not reproduce the image as effectively. The uneven distribution of ink on the typewriter characters aids in shape matching, compared to the perfectly flat distribution across the characters of the font. Crucially, even the darkest typewriter characters are less than fully black, which yields a greater tonal range when characters are allowed to overlap.

The center and right columns in Figure 20 exhibit progressively better tonal range as the character set includes a better distribution of grays. Even if we limit ourselves to two intensities, results improve by having medium and light intensities, rather than dark and medium. There is a trade-off here: overlapping lighter characters can produce a smoother tonal range, at the expense of requiring more overlaps to produce full black.

The character set used for the majority of this study is shown in Figure 19. For aesthetic effect (or ease of typing), a subset of the characters can be specified; since we were pursuing the closest match to the target image, we used the full set.

## 5.6 Timing notes

The algorithm is expensive due to the large number of image compositing and comparison operations required to determine the best character selection for a given position, multiplied by the number of iterations before convergence. Each of these operations is performed at a high resolution (around 1000 pixels total) at 8-bit depth.

We leaned on Google Colaboratory, a cloud computer service that supplies 2 threads of an Intel Xeon processor at 2.3GHz and 13GB of RAM [3]. The software is written in Python 3, making use of numpy, OpenCV and scikit-image libraries.

The top left image in Figure 5 is composed of two passes of 2670 characters each. To converge to a stable state the algorithm visited each character position 25 times. The first pass took 29 minutes to complete; subsequent overstrike passes converge more quickly. Auto-alignment took an additional 10 minutes. In total, the image took approximately one hour to generate. These timings are representative and vary little over different input photographs.

Convergence slows down as resolution is increased, even accounting for the larger number of characters: at 420 characters, the algorithm took 0.64 seconds per character position; at 2670 characters, 0.77 seconds; and at 6392 characters, 0.94 seconds.

## 5.7 Limitations

### 5.7.1 Algorithmic limitations

The most notable limitation of this method is that the optimization is quite time-consuming. We were primarily concerned with image quality; future work could focus on acceleration.

Halftoning-type patterns can emerge, which falsely delineate tonally consistent areas, creating spurious shapes. Repeated characters create false textures. Both of these can be addressed by penalizing repeated adjacent characters and/or through dithering. For large images, dither would also subjectively improve tone matching over the highlight areas [6, 14].

Our implementation of overstrike was made for pragmatic reasons; in it, each pass is computed in sequence with no backtracking, so selections made in the first pass are not revisited. For example, two slash characters ( \ and / ) typed in the same position might provide a better shape match than the typewriter's serif X character, but this might only be realized if both overstrike layers are considered at once.

### 5.7.2 Physical limitations

The algorithm is sufficiently optimized that human error in typing is a noticeable limitation. It is especially difficult to use precise, light strike force. While the addition of softly typed characters into the character set is crucial for producing smooth midtones and highlights without employing dithering, physical limitations make reliable reproduction of the lightest tones difficult, as seen in Figure 21. When typing an image generated by the algorithm, human error increases due to frequent switching between levels of strike force.

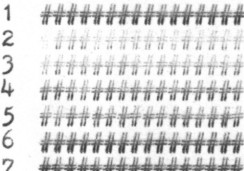

Figure 21: Variation in strike force. Top to bottom: Intentional variation; Left to right: Unintentional variation

A manual solution could be to arrange the typing instructions so that the typist will first type all "hard" characters, then fill in the "soft" characters. The typist could then apply consistent strike force throughout each pass, lowering the difficulty. Ultimately, an automated typewriter would provide better consistency in strike force.

The typewriter itself is also a source of error. The distribution of ink on the ribbon is not perfectly uniform, so even with precise strike force, the scanned characters may not be exactly reproduced when typed later. The typewriter may also drift out of alignment due to variation in the amount the platen is moved or rotated at each keystroke.

These issues can be ameliorated somewhat by improved preparation of the character set. Multiple typed copies of each character could be averaged both in intensity and alignment. This would reduce tone error due to variation in strike force and ribbon inkiness, and increase placement consistency between the simulated and physical typed output.

## 6  CONCLUSIONS AND FUTURE WORK

The typewriter, as an analog mechanical device, possesses degrees of freedom not readily available to digital ASCII art: multiple characters can be typed at a single grid location; typing is not restricted to a grid; and characters can be typed with different levels of force. Our method is able to produce high quality output, both physical (by manually typing the selected characters) and digital (displaying the simulated typed result).

We used four overlapping layers of characters as a compromise balancing mechanical reproducibility (can the image be physically typed?) and image quality (does it faithfully represent the input?). We were able to obtain good results over a variety of subject matter under these conditions; dark tones and fine details are handled with overlapping and overstriking, while application of lighter strike force helps increase fidelity in light areas of the image.

We suggest a few different directions for future work, both in modifying details of the existing algorithm and in extending automatic typewriter art to handle additional degrees of freedom.

As the choice of error metric has a strong effect on the result, other metrics should be explored. To improve shape matching, we could employ Alignment Insensitive Shape Similarity, created for ASCII art. To produce results with good tone matching, it would have to be used in combination with other metrics. Machine optimization of loss function blending hyperparameters could improve results. As previously noted, we do not want to perfectly reproduce the original; future work could involve reframing the error function so as to highlight the typed characters as well as the original image.

Freehand typewriter artists, such as Paul Smith, take advantage of rotating the page: characters can be typed at any angle [19]. We would like to investigate allowing some layers to be typed with the paper rotated.

We explored only static images in this paper. It would be interesting to investigate applications to video. As with other non-photorealistic rendering techniques, temporal coherence will be an issue. Such a method could consider character similarity between frames, and would probably benefit from allowing characters to move freely, i.e., not restricted to the grid; there would be less expectation of typing out a video sequence, so mechanical reproducibility would not be as strong a consideration.

Finally, this work has focused on achieving fidelity by optimizing the algorithm more than optimizing physical reproduction. The algorithm can be adapted to increase similarity between rendered and typed results: improved compositing would render a physically realistic black level, and the algorithm could be made to discourage reliance on precise character placement for tone matching, including by adding blur or reducing resolution during selection. The physical results could be improved by exploring human factors or robotics in future work.

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
