# OpenReview forum: "Algorithmic Typewriter Art: Can 1000 Words Paint a Picture?"
_graphicsinterface.org/Graphics_Interface/2021/Conference — GI 2021_

### Official Review · AnonReviewer2 · 2020-12-29
**Review of "Algorithmic Typewriter Art: Can 1000 Words Paint a Picture?"**

**Rating:** 5
**Confidence:** 4

**Review:**

This paper presents a method for approximating  a source image using a mechanical typewriter's character set.  A scanned set of typed characters provides tiles able to contribute to the tone of an output image.  Given a reference image as input, the paper uses simulated annealing to choose scanned glyphs in a few overlapping grids in order to best approximate the tone and structure of the image.  The process can be further modulated by allowing overstrike (multiple glyphs in the same position) and variable striking force.  The paper presents a number of results—most rendered synthetically, and a couple typed manually.

At first glance, this is a perfectly reasonable idea, and it produces results that are fun and attractive.  It's pretty classic NPR: we're processing a source image to create an artistic output, but the output medium is constrained and we have to produce the best possible result under those constraints.  The algorithm, then, is a kind of projection operator onto the manifold of all images that can be formed from type.  There's not a huge amount of novelty here, but it's still interesting enough.

Because the technique operates on scanned images of typed characters, I see a kinship with Photomosaics, something not mentioned in the paper.  (The use of multiple overlapping grids is the obvious departure from Photomosaics.)  There's also some connection to Hertzmann's 1998 Painterly Rendering paper, in that we proceed through a set of layers, each layer optimized to pick up the residual energy not handled by previous layers.  I'd swear I've also seen the idea of an asymmetric MSE somewhere before, but I'm not sure I'll be able to pull that out of the memory hole.

However, there were a number of small-to-medium issues with the paper, which left me unsure about whether it should be accepted.  Here are my main concerns:

 * There's a tension in work like this, having to do with the optimization's objective function.  The problem is that if you somehow achieve a perfect solution (i.e., no error between the source image and the typed characters), you won't be happy—you'll just have the original image again!  These artworks are interesting precisely because there's error in the output, allowing the individual typed characters to show through while still communicating the source image.

   I think the paper should do more to acknowledge this tension.  Perhaps it's as simple as noting in the introduction that we'll always end up with error in the optimization, and that's a good thing.  But this could inform changes to the algorithm to favour individual characters showing through when the image approximation is good enough.

 * It should always be clear which figures are rendered and which are physically typed.  One option is to say something like "unless otherwise noted, all images are computer-generated" (I think they're the majority...).

 * Figure 1: How many characters are there?  What does "12 layers in 4 positions" mean?  That's not like anything described in the paper.

 * The introduction should clarify that we're talking about mechanical typewriters—even early electric typewriters can't vary their strike force.

 * By no means does Figure 3 outline an algorithm.  It shows the inputs and outputs, and nothing else.  I'm not sure how useful the figure is; I'd like to see more of the scanned typed page.

 * The connection made to halftoning in Section 2 is tenuous.  The scanned type samples are most definitely variable in tone, and Figure 17 shows clearly that this makes a big difference.  I'm not sure what to do about that.

 * There's some preprocessing that's not discussed in the paper.  How is a scanned page of type decomposed into individual glyphs?  How are the metrics of the typewriter (character spacing, leading, etc.) determined?  Obviously these problems are interrelated.

 * How does the complexity of the algorithm vary with the number of distinct characters available?  Are all of a typewriters characters used?  If we render a digital font, which characters are chosen?

 * What is up with Algorithm 1?  It's never referenced in the paper, and its details directly contradict those provided in the surrounding text.  The initialization is different, as is the order in which character positions are visited.  It doesn't use simulated annealing.  And so on.

 * The algorithmic details in Section 3 are confusing.  Some pieces add up to a complete algorithm, but some pieces supersede earlier ones.  The description of simulated annealing overrides the stopping condition in Section 3.2, right?

 * The cropping in Section 3.4 isn't described in nearly enough detail for me to understand what's going on.

 * Section 3.5 describes simulated annealing in general terms, but doesn't provide enough detail to reproduce the probabilities and cooling schedule used the produce the results in the paper.  The paper should provide precise details about temperatures and acceptance probabilities, even if it's only in an appendix or in supplemental material.

 * Section 4 is called "Results", but to my eye it contains a lot more information and algorithmic details.  I think a lot of this material should be reorganized to pull out a real results section.

 * 4.2.1: So, do we start with a blank page, or with random characters, as stated in Algorithm 1?

 * I don't understand how to interpret Figure 7.  I need more guidance on what this plot means.  I also have no idea what the labels on the data points refer to.

 * 4.2.2 is slightly confusing.  The text says "SA consistently finds a lower-error state...", but the graphs in Figure 6 show PSNR, where higher is better.

 * In Section 4.2.3, I'm surprised that there's no attempt to experiment with light-to-dark or dark-to-light selection orders.

 * Section 4.3.1: Again, I see that cropping is important, but we're given no information on how it's done.  Also, give error measurements for the images in Figure 10.

 * Section 4.5 says that less tonal variation can "draw more attention to the textual characters".  But as noted above, maybe that's a good thing?

 * In Section 4.6, how are overlapping letters composited?  Scanned letters are greyscale images and not masks, so you need to something to combine overlapping layers of ink.  What is it?  Is it physically realistic?

 * The magic number of 5790, discussed in Section 5.1, is cute but tenuous.  After all, no attempt is made to use "words" (which would be very interesting, of course).

 * The experiment with multiple strike forces is interesting, but ultimately I'm not sure it's successful.  It's an idea that works reasonably well with synthetic results, but we see evidence that it's too fussy to reproduce by hand.  Are there examples of hand-made typographic art that exploits multiple strike forces?

 * Please provide links to source photos and scanned character sets, for anybody interested in following up on this work.  This could be in a supplement.

 * Section 5.2 says "Tonally, the gradient produced in the rendered result is reflected in the physical typed result".  Where is this?  What figure am I looking at?

 * The hand-typed result in Figure 20/21 is great!

 * Figure 22 and the text underneath it need a lot of work.  Text like "the variation in strike force is apparent on B and % characters" and "note how the 12 and y characters" is almost useless, especially when these characters appear multiple times.  If there's something specific to look at in Figure 22, point to it or highlight it in the images, possibly by adding close-ups.

   Of course, it would be much easier to demonstrate the variation in tone in a simple experiment: just type the same letter a lot of times and show it to us.  Maybe measure the variation directly.  There's no need to evaluate this variation only in-situ in a complex image.

 * What's 0.15% in Section 5.2?  Maybe just say how many characters were mistyped?  How was that measured?

 * Overall the writing is fine, though there are grammatical mistakes in some places.  For example: "this partly caused" in Section 5.2, and "it is more probably" in Section 3.5".  I'm sure these issues can be eliminated with careful proofreading.

Adding up all the concerns above, I'm left somewhere in the middle.  The technique is probably publishable, but I think that many improvements are needed to the paper, perhaps too many to accept it directly.

Postscript: if you haven't seen it, be sure to watch the 1988 film "Primiti Too Taa", my favourite piece of typewriter art. https://www.youtube.com/watch?v=mOdsmfjCunM

---

### Official Review · AnonReviewer3 · 2021-01-10
**The paper presents and analyses an approach to generate Typewriter Art. Overall, results are good and the paper is clearly written. Some limitations of the approach could be discussed in more detail.**

**Rating:** 8
**Confidence:** 4

**Review:**

The paper presents an algorithm to convert images into Typewriter Art. This leads to a problem formulation that is similar to ASCII art, but extended to account for the nuances of this medium (e.g. key strikes of different strength, and the ability to type multiple layers of characters on the same page). To search the space of possible outputs for a high fidelity match to the original image, the author’s propose two algorithms: one taking a greedy approach, and the other using simulated annealing.

Although the application is a bit niche, typewriter art provides an interesting extension of previous work on ASCII art. The presented results analysing how various loss functions, resolutions, and the character set and number of overstrike characters affect results is reasonably through and demonstrate how algorithm parameters and design decisions affect the aesthetic and fidelity of results.

The ability to approximately reproduce the typewriter art using a physical typewriter is charming, although I expected a bit more discussion on how to simulate and account for physical limitations of the actual device algorithmically.

On the negative side, the algorithm runtimes are quite long (~1 hour for 6k charachters). A brief discussion of how this could be improved would be useful (it seems like some amount of parallelization should be possible). I’m also curious about the runtime of the greedy algorithm compared to simulated annealing.

From an NPR perspective, I did find it a bit strange that most of the results and discussion focused on creating a high fidelity approximation of the original image as opposed to recreating the unique aesthetic of typewriter art. I was also curious if the authors considered animated typewriter art as a future work (something that seems very difficult to achieve using an actual typewriter).

In summary, as a first foray in this area, the paper does a good job of framing the problem, providing a workable approach and highlighting many open problems. As indicated above, I do feel that further discussion of how some of these open problems could be addressed would be helpful and add to the value of the paper.

Minor comments:
Fig. 11: I’m assuming left is un-optimized and right is optimized?
Fig. 16: I can’t find a reference to this figure in the text.

---

### Official Review · AnonReviewer1 · 2021-01-13
**This study explores the effectiveness of typewriter art that deploys a range of techniques to reproduce a given input image. The results are interesting, although, in the grand scheme of things,  I'm not sure how much value there is in comparing erroneous manually produced examples.**

**Rating:** 7
**Confidence:** 3

**Review:**

This is a well written study that presents a novel technique to generate typewriter art equivalent images based on a given example image. The topic of study is a little bit abstract, and some may question the value or potential contribution of such work, however, the research presented here seems sound and this may be of interest to some researchers.

There are some weaknesses in the work though, for example, the algorithm presented at the start of Section 3.1 seems to be overly simplified, to the extent that it is not really possible to understand what the claimed technique is actually doing when it processes and produces outputs. The outputs themselves are impressive and the technique (if it were revealed in its full glory) may well be of value to a range of researchers within this field and possibly in related fields of study.

The manually produced equivalent outputs that are used to evaluate the technique are interesting, but it highlights the fact that the instructions produced by this technique must be overly complex and extremely difficult to follow if so many errors occurred when attempting to follow them. It's not clear what the value of this element of the study was in the grand scheme of things, and if human error is creeping into the process, then perhaps more effort should be invested into ensuring the instructions are delivered in a comprehensible format. Indeed there does not seem to be any example of what the instructions output by the technique look like etc. Overall, I believe there is scope to talk about the limitations of this study in a bit more depth and also to deliver amore formal representation of the technique/s that have been employed by the system, however, it is a well written, and rigorous study, that I believe makes a reasonable contribution to the Graphics Interface community.

---

### Meta-Review · Area_Chair1 · 2021-01-14

**Recommendation:** Accept
**Confidence:** 3

**Metareview:**

Although a little niche, this is an interesting and well presented study which makes some useful and novel contributions to the GI community. A more detailed presentation of the algorithm at the start of Section 3 and a more considered limitations and future research section along with a more thorough exploration of how some of the open problems presented may be addressed would strengthen the work further. The comprehensive list of 'small to medium issues' identified by Reviewer #2 together represent a significant collection of issues, which if considered and accommodated, would without doubt strengthen this submission considerably - the reviewer should be commended for carrying out such a thorough and helpful review. We trust that the authors will recognise the immense value of receiving such detailed and helpful feedback and that they will take these extremely useful comments forward with their research. All of this considered, I believe this work is of sufficient value to warrant acceptance.

---

### Decision · Program_Chairs · 2021-01-16

Accept